# A Phase Field Study of the Influence of External Loading on the Dynamics of Martensitic Phase Transformation

**DOI:** 10.3390/ma16216849

**Published:** 2023-10-25

**Authors:** Genggen Liu, Jiao Man, Bin Yang, Qingtian Wang, Juncheng Wang

**Affiliations:** School of Mechanical Engineering, Xinjiang University, Urumqi 830054, China; liugg1345@163.com (G.L.);

**Keywords:** phase field simulation, martensitic phase transformation, phase transition kinetics, martensitic variant, phase field model

## Abstract

An elastoplastic phase field model was employed for simulations to investigate the influence of external loading on the martensitic phase transformation kinetics in steel. The phase field model incorporates external loading and plastic deformation. During the simulation process, the authenticity of the phase field model is ensured by introducing the relevant physical parameters and comparing them with experimental data. During the calculations, loads of various magnitudes and loading conditions were considered. An analysis and discussion were conducted concerning the volume fraction and phase transition temperature during the phase transformation process. The simulation results prominently illustrate the preferential orientation of variants under different loading conditions. This model can be applied to the qualitative phase transition evolution of Fe-Ni alloys, and the crystallographic parameters adhere to the volume expansion effect. It is concluded that uniaxial loading promotes martensitic phase transformation, while triaxial compressive loading inhibits it. From a dynamic perspective, it is demonstrated that external uniaxial loading accelerates the kinetics of martensitic phase transformation, with uniaxial compression being more effective in accelerating the phase transformation process than uniaxial tension. When compared to experimental data, the simulation results provide evidence that under the influence of external loading, the martensitic phase transformation is significantly influenced by the applied load, with the impact of external loading being more significant than that of plastic effects.

## 1. Introduction

The martensitic phase exhibits ideal mechanical performance; its microstructure and composition determine the mechanical properties of steel, making it one of the crucial constituent phases in high-strength steel. The high strength and hardness of the martensitic phase can be attributed to the solid solution strengthening by carbon atoms and the complex martensitic microstructure formed through rapid, diffusionless phase transformation. Consequently, many scholars have conducted in-depth experimental and theoretical research on the microstructure of martensite and the mechanical properties of martensitic steels [1,2,3,4]. In order to gain a profound understanding of the relationship between microstructure and properties in steel, it is imperative to delve into the processes of martensitic phase transformation under various complex conditions and the evolution of martensitic microstructures.

At present, phase field methods, as a powerful tool for predicting microstructural evolution, are widely applied in materials’ solidification [5] and solid-state phase transformations [6,7]. Especially in martensitic phase transformations, refined theories and various phase field models can accurately predict the evolution of microstructures during the martensitic phase transformation process. In this study, numerical simulation methods [8] have been employed to investigate martensitic phase transformations. In engineering applications, there are often situations where experiments are impractical. Numerical simulation, with its specific computational techniques, can replicate complex processes. Compared to experimental research, numerical simulation methods offer advantages such as cost-effectiveness, the ability to simulate conditions that are not achievable through experimental means, and comprehensive data collection. However, in the current state of numerical simulation, simplifications are often made to boundary conditions and material properties during the simulation and analysis process. The analysis results can significantly impact structural discretization, leading to varying outcomes and precision levels. The model employed in this study applies to the qualitative phase transition evolution of Fe-Ni alloys, as the crystallographic parameters conform to the volume expansion effect. Using Fe-Ni alloys as the prototype, we contrast experimental data with simulations to elucidate the mechanisms of external loading on martensitic phase transformation. However, due to variations in alloy parameters, the effects may differ, primarily concerning the crystallographic parameters of the martensitic phase transformation. The phase transition evolution is associated with its chemical free energy parameters, such as the Landau free energy coefficient ($a_{0}$) and the phase transition latent heat (Q). The influence of tensile and compressive effects on phase transformation is related to crystallographic parameters. The numerical simulation of martensitic phase transformations originated from the pioneering work of scholars such as Khachaturyan and Wang [9]. Building upon the foundation of micromechanics theory and inclusion physics, Khachaturyan and his colleagues proposed a phase field model for martensitic phase transformations, known as the phase field micromechanical model for martensitic phase transformations. In this model, a time-dependent Ginzburg–Landau (TDGL) equation is employed to simulate the evolution of martensite in single crystals. Artemev and his colleagues introduced a three-dimensional phase field model based on phase field microelasticity theory to investigate the influence of applied stress on the cubic-to-tetragonal martensitic phase transformation [10]. Yamanaka and other researchers [11] used an elastoplastic phase field model to study the effects of plastic deformation on martensitic phase transformations. Subsequently, Yeddu and colleagues built upon Khachaturyan’s elastoplastic phase field microelasticity theory, developing a phase field model that incorporates plastic deformation and anisotropic properties. This model was used to investigate autocatalysis in martensitic phase transformations and classical features of martensitic microstructures [12]. Currently, the phase field method based on microelasticity theory has been demonstrated to effectively simulate martensitic phase transformations under various complex conditions in single crystals.

Stress-assisted martensitic phase transformations are commonly observed in high-strength steel materials. They find extensive applications because they can enhance the mechanical properties of materials by forming different martensitic structures under external loads. While numerous scholars have conducted extensive research on stress-assisted martensitic phase transformations [13,14], there have been relatively few reports on using phase field methods to simulate martensitic phase transformations under different external loading conditions, specifically to investigate the associated dynamic characteristics of phase transformations. Therefore, further in-depth research is needed to explore the microstructural evolution, phase transformation processes, and phase transformation temperatures under different external loading states.

This study employed a phase field method to conduct an in-depth investigation into the martensitic phase transformation of Fe-Ni single crystals under the influence of external loading conditions. This study employed a coupled elastic–plastic phase field model incorporating external loads to simulate martensitic phase transformation under various conditions, including no external load, uniaxial loading, and triaxial compression loading. The study conducted an analysis and discussion on the evolution of microstructures of variants during the phase transformation process, as well as changes in volume fractions. At the same time, this study compared the effects of external loads and plasticity on martensitic phase transformation. Statistical analysis was performed on the data related to elastic strain energy and equivalent plastic strain during the phase transformation process. Researchers such as Patel and Hagiwara [15,16] have experimentally investigated the martensitic start temperature (*Ms*) changes and martensite volume fraction in Fe-Ni alloys under different external loading conditions. In this study, our phase field model successfully predicted the trend of *M_s_* temperature changes. This not only provides a reference for understanding the mechanism of external loading effects on martensitic phase transformations but also plays a significant role in controlling martensitic phase transformations and gaining deeper insights into the process of martensitic phase transformations.

## 2. The Elastoplastic Phase Field Model

The microstructure formed during martensitic phase transformations can be described using a set of long-range order (LRO) parameters, which represent changes in crystal symmetry during the phase transformation process. The phase field model employed in this paper is constructed based on the research of Wang, Chen, Yeddu, and others [9,12,17]. In solid-state phase transformations, the Gibbs free energy (G) is defined as the sum of three energies, namely, the chemical free energy ($G_{chem}$), gradient energy ($G_{grad}$), and elastic strain energy ($G_{el}$).
(1)$$G=G_{chem}+G_{grad}+G_{el}$$

The system’s Gibbs free energy governs the evolution of martensitic microstructures. For diffusionless phase transformations, the time-dependent Ginzburg–Landau (TDGL) dynamic equation, as proposed by Allen-Cahn, can be employed to describe this process:(2)$$\frac{{\partial \eta }_{p}\left(r,t\right)}{\partial t}=-\sum_{q=1}^{n}L_{pq}\frac{\delta G}{\delta \eta_{p}(r,t)}$$

Here, *r(x,y,z)* is a vector representing spatial coordinates in Cartesian coordinates, η is the long-range order parameter or phase field variable, with the austenite phase being 0 and martensite phase being 1 in this model. $L_{pq}$ represents the dynamic coefficient indicating interface mobility and is assumed to be isotropic. In this context, *n* equals 3, signifying that typical martensitic microstructures in Fe-Ni alloys can exhibit three different orientations or variants.

In the context of martensitic phase transformations, for the local free energy density, the Landau free energy is typically defined by a fourth-order Landau polynomial [18]:(3)$$f\left(\eta_{p}\right)=\frac{A}{2}\sum_{p=1}^{n}\eta_{p}^{2}-\frac{B}{3}\sum_{p=1}^{n}\eta_{p}^{3}+\frac{C}{4}\sum_{p=1}^{n}\eta_{p}^{4}+\frac{D}{2}\sum_{p=1}^{n}\eta_{p}^{2}(\sum_{q=1}^{n,q\neq p}\eta_{q}^{2})$$

The chemical driving force and the energy barriers between different phases or variants determine the coefficients *A*, *B*, *C*, and *D*. Since this type of function does not explicitly represent thermodynamic or physical variables, this study refers to the work of Tae Wook Heo and Long-Qing Chen [19], which explicitly combines temperature and latent heat for the transformation. In other words, the function of chemical free energy is as follows:(4)$$G_{chem}=\int f\left(\eta_{p},T\right)dV\;\;\;\;\;\;\;\;\;\;\;\;\;\;\;\;\;\;\;\;\;\;\;\;\;=\int \left(\left[a_{0}+\frac{3Q\cdot \left(T-T_{0}\right)}{T_{0}}\right]\cdot \sum_{p}\eta_{p}^{2}-\left[{2a}_{0}+\frac{2Q\cdot \left(T-T_{0}\right)}{T_{0}}\right]\cdot \sum_{p}\eta_{p}^{3}+a_{0}\cdot \sum_{p}\eta_{p}^{4}\right)dV$$

Here, T represents the undercooling temperature, $T_{0}$ is the stress-free equilibrium temperature, and $a_{0}$ is an empirical parameter. The dimensionless local free energy as a function of the order parameter is schematically illustrated in Figure 1.

Figure 1, in the dimensionless local free energy schematic, clearly indicates that the phase transition proceeds in the direction of lower energy. Temperature values in Figure 1 are specified as T = 260 K, 280 K, 300 K, 320 K, and 340 K, with $T_{0}$ = 405 K.

The gradient energy is defined as the sum of energy contributions arising from the non-uniformity of the order parameter, as shown below [20]:(5)$$G_{grad}=\int \left[\frac{1}{2}\sum_{p=1}^{n}\beta_{ij}(p)\nabla_{i}\eta_{p}\nabla_{i}\eta_{p}\right]dV$$
where $\beta_{ij}$ is the gradient coefficient matrix determined by the interfacial energy and interface width. In this study, it is assumed that the interface properties are isotropic.

The elastic strain energy for a mixture system with arbitrary parent and martensite phases is given by [21]
(6)$$G_{el}=\frac{1}{2}\int C_{ijkl}\varepsilon_{ij}^{el}(r)\varepsilon_{kl}^{el}(r)dV$$
where $C_{ijkl}$ is the fourth-order tensor of elastic constants, and $\varepsilon_{ij}^{el}(r)$ represents the elastic strain tensor. The elastic strain tensor $\varepsilon_{ij}^{el}(r)$ is defined as the difference between the total strain tensor $\varepsilon_{ij}^{tot}(r)$ and the eigenstrain tensor $\varepsilon_{ij}^{0}(r)$:(7)$$\varepsilon_{ij}^{el}(r)=\varepsilon_{ij}^{tot}(r)-\varepsilon_{ij}^{0}(r)$$

The total strain tensor $\varepsilon_{ij}^{tot}(r)$ is defined as the sum of the homogeneous strain tensor ${\bar{\varepsilon }}_{ij}^{tot}(r)$ and the inhomogeneous strain tensor ${\delta \varepsilon }_{ij}^{tot}(r)$:(8)$$\varepsilon_{ij}^{tot}(r)={\bar{\varepsilon }}_{ij}^{tot}(r)+{\delta \varepsilon }_{ij}^{tot}(r)$$

When the macroscopic shape of the system remains fixed during the phase transformation process, the homogeneous strain tensor ${\bar{\varepsilon }}_{ij}^{tot}(r)$ is defined as follows:(9)$${\bar{\varepsilon }}_{ij}^{tot}(r)=0$$

The inhomogeneous strain is defined as the deviation from the homogeneous strain and does not affect the macroscopic deformation. The inhomogeneous strain is represented by the elastic displacement field $u_{i}(r)$:(10)$${\delta \varepsilon }_{ij}^{tot}\left(r\right)=\frac{1}{2}(\frac{\partial u_{i}}{\partial r_{j}}+\frac{\partial u_{j}}{\partial r_{i}})$$

Assuming mechanical equilibrium is reached, the elastic solution is obtained by solving the following mechanical equilibrium equations:(11)$$\nabla_{j}\sigma_{ij}=\nabla_{j}\left[C_{ijkl}\cdot \left({\bar{\varepsilon }}_{kl}^{tot}\left(r\right)+{\delta \varepsilon }_{kl}^{tot}\left(r\right)-\varepsilon_{kl}^{0}(r)\right)\right]=0$$

The local stress $\sigma_{ij}$ in the equation is calculated using Hooke’s law. To describe the elastoplastic deformation leading to martensitic phase transformations, plastic strain is introduced in the model. The eigenstrain is defined as the sum of phase transformation strain $\varepsilon_{ij}^{t}(r)$ and plastic strain $\varepsilon_{ij}^{p}(r):$(12)$$\varepsilon_{ij}^{0}\left(r\right)=\varepsilon_{ij}^{t}\left(r\right)+\varepsilon_{ij}^{p}(r)$$

According to the microelasticity theory, the phase transformation strain $\varepsilon_{ij}^{t}(r)$ is a linear combination of the stress-free phase transformation strain $\varepsilon_{ij}^{00}(p)$ and the corresponding phase field variable $\eta_{p}$:(13)$$\varepsilon_{ij}^{t}\left(r\right)=\sum_{p=1}^{n}\eta_{p}(r,t)\varepsilon_{ij}^{00}(p)$$

Here, $\varepsilon_{ij}^{00}(p)$ is the stress-free phase transformation strain of a variant, calculated based on lattice parameters and the orientation relationship between the parent phase and martensite phase [22]:(14)$$\varepsilon_{ij}^{00}\left(p\right)=\left(F_{p}^{T}F_{p}-I\right)/2$$
where $F_{p}$ is the deformation gradient tensor for the martensitic phase transformation. The deformation gradient tensor, $F_{p}$, is computed based on relevant crystallographic representation theory, utilizing the Bain strain for the transformation from a cubic phase to a tetragonal phase, and the stress-free phase transformation strain $\varepsilon_{ij}^{00}(p)$ can be given by the Bain strain as follows:(15)$$\varepsilon_{ij}^{00}\left(1\right)=\left[\begin{array}{ccc} \varepsilon_{3} & 0 & 0\\ 0 & \varepsilon_{1} & 0\\ 0 & 0 & \varepsilon_{1} \end{array}\right]\varepsilon_{ij}^{00}\left(2\right)=\left[\begin{array}{ccc} \varepsilon_{1} & 0 & 0\\ 0 & \varepsilon_{3} & 0\\ 0 & 0 & \varepsilon_{1} \end{array}\right]\varepsilon_{ij}^{00}\left(3\right)=\left[\begin{array}{ccc} \varepsilon_{1} & 0 & 0\\ 0 & \varepsilon_{1} & 0\\ 0 & 0 & \varepsilon_{3} \end{array}\right]$$
where $\varepsilon_{1}=\left(a_{bcc}-\frac{\sqrt{2}}{2}a_{fcc}\right)/\frac{\sqrt{2}}{2}a_{fcc}$ and $\varepsilon_{3}=\left(c_{bcc}-a_{fcc}\right)/a_{fcc}$, where $\varepsilon_{1}$ and $\varepsilon_{3}$ are related to the lattice parameters of the parent phase $\left(a_{fcc}\right)$ and martensitic phase $\left(a_{bcc},c_{bcc}\right)$. Based on the above derivation, the elastic strain energy can be rewritten using Equation (6):(16)$$G_{el}=\frac{1}{2}\int C_{ijkl}\left(\frac{1}{2}\varepsilon_{ij}^{tot}\left(r\right)\varepsilon_{kl}^{tot}\left(r\right)-\varepsilon_{ij}^{tot}\left(r\right)\varepsilon_{kl}^{0}\left(r\right)+\frac{1}{2}\varepsilon_{ij}^{0}(r)\varepsilon_{kl}^{0}(r)\right)dV$$

When the local von Mises stress reaches the yield stress $\sigma_{y}$, the material begins to undergo plastic deformation. The yield criterion can be determined using the von Mises yield criterion, which can be expressed by the following equation to determine if the equivalent stress $\sigma_{eq}$ has reached the yield limit:(17)$$\sigma_{eq}^{2}-\sigma_{y}^{2}=\frac{1}{2}{\left(\sigma_{xx}-\sigma_{yy}\right)}^{2}+\frac{1}{2}{\left(\sigma_{yy}-\sigma_{zz}\right)}^{2}+\frac{1}{2}{\left(\sigma_{xx}-\sigma_{zz}\right)}^{2}+\;3\left(\sigma_{xy}^{2}+\sigma_{yz}^{2}+\sigma_{xz}^{2}\right)-\sigma_{y}^{2}\geq 0$$
where $\sigma_{eq}$ represents the von Mises equivalent stress, and $\sigma_{y}$ represents the yield stress. In order to model martensitic phase transformations under external loading conditions, the additional Gibbs energy induced by externally applied stresses needs to be included in the system’s Gibbs free energy, as shown in [23].
(18)$$G=G_{chem}+G_{grad}+G_{el}+G_{appl}$$
where $G_{chem}$, $G_{grad}$, and $G_{el}$ are the chemical, gradient, and elastic parts of the Gibbs free energy. $G_{appl}$ is the additional Gibbs free energy induced by externally applied stresses. Therefore, $G_{appl}$ can be expressed as follows:(19)$$G_{appl}=-\sigma_{ij}^{appl}\varepsilon_{ij}^{t}\left(r\right)$$
where $\sigma_{ij}^{appl}$ is the externally applied stress tensor, represented by the Cauchy stress tensor as follows:(20)$$\sigma_{ij}^{appl}=\left[\begin{array}{ccc} \sigma_{xx} & \sigma_{xy} & \sigma_{xz}\\ \sigma_{yx} & \sigma_{yy} & \sigma_{yz}\\ \sigma_{zx} & \sigma_{zy} & \sigma_{zz} \end{array}\right]$$

From Equation (19), it can be observed that the applied stress affects the phase transformation strain $\varepsilon_{ij}^{t}\left(r\right)$, which in turn influences the eigenstrain $\varepsilon_{ij}^{0}\left(r\right)$ in the elastic strain energy $G_{el}$ in Equation (18).

## 3. Simulation Parameters and Conditions

In this study, physical parameters for Fe-Ni alloys were selected based on [19,24]. The relevant physical parameters are shown in Table 1.

In the phase field calculations, a semi-implicit Fourier spectral method proposed by Chen [17] was used to solve the dynamic equations. The simulation system had a grid size of 65 × 65 × 65. To study the growth process of martensitic phase transformations, a small cubic martensite nucleus with a side length of 1.6 μm was assumed to pre-exist at the center of the simulation domain. A single crystal grain undergoing martensitic phase transformation was considered the simulation region and had a physical size of approximately 16 μm. The iso-surface of the phase field variable (g = 0.7) is shown in all figures. For simplicity, Equation (7) in this paper is solved in dimensionless form, with dimensionless parameters as listed in Table 2.

Here, $Q^{*}$ and $a_{0}^{*}$ are dimensionless parameters for the phase transition latent heat and the Landau free energy coefficient, obtained by dividing these quantities by a characteristic energy, where the characteristic energy is represented as E_0_ = 1.026 × 10^7^. The dimensionless time ∆t* = 0.01 corresponds to a real time of approximately 0.975 nanoseconds. In the study, the parameter *t^*^* represents non-dimensional time, and the colors of the phase field variable, such as red, blue, and green, correspond to martensitic variants 1, 2, and 3, respectively. The model employs stress boundary conditions to solve mechanical problems. External stresses are selected so that they remain smaller than the yield stresses of both the austenite and martensite phases. Different simulations were conducted under various loading conditions in the phase field simulations presented in this paper. To compare with relevant experimental results and validate them, uniaxial tensile and uniaxial compressive loads of 100 MPa, 130 MPa, and 150 MPa were applied along the [100] direction. The triaxial compressive loading conditions correspond to isotropic hydrostatic pressure, where equal pressure is applied from all sides. To match the experimental conditions in the reference literature, triaxial compressive loads of 100 MPa, 130 MPa, and 150 MPa were used. The phase field model was implemented as code in Matlab R2021b.

## 4. Simulation Results and Analysis

### 4.1. Martensitic Phase Transformation without External Loading

The evolution of martensite volume fraction during martensitic phase transformation without external loading is shown in Figure 2. The three-dimensional microstructure at t* = 0, t* = 35, t* = 55, and t* = 65 is depicted in the figure.

From Figure 2, it can be observed that the martensite volume fraction increases with time, and it increases rapidly after t* = 50. From the three-dimensional microstructure images, it is evident that as the phase transformation progresses, the small cubic martensite nucleus pre-existing at the center of the simulation domain gradually transforms into different martensite variants. Growing martensite can induce other unstable martensite nucleus embryos to transform into stable nuclei and start growing, a phenomenon known as self-catalytic nucleation [25]. Martensite nuclei create stress fields around them during growth, and to reach a stable state, nucleation of new variants is required to reduce strain energy. This evolutionary process reduces the stress generated around growing martensite due to self-catalytic nucleation.

### 4.2. Microstructure Evolution of Martensitic Phase Transformation under External Loading

The microstructure obtained under uniaxial tensile loading with σ = 100 MPa is shown in Figure 3.

As depicted in the figure, the final microstructure is primarily composed of two variants, variant 2 and variant 3. The evolution is driven by the energy imposed externally, aiming to minimize the Gibbs free energy, which means the evolution proceeds in the direction of lower energy. According to the phase field model, variant 1 is controlled by the Bain strain tensor, which compresses along the [100] direction but stretches along the [010] and [001] direction. Under uniaxial tensile loading along the [100] direction, $G_{appl}$ increases, leading to the suppression of variant 1. However, variants 2 and 3 controlled by the Bain strain tensor experience a decrease in $G_{appl}$ under uniaxial tensile loading along the [100] direction. Therefore, the formation of variants 2 and 3 is promoted.

The microstructure obtained under uniaxial compressive loading with σ = −100 MPa is shown in Figure 4. As seen in Figure 4, under uniaxial compressive loading, the formation of variant 1 is favorable. In contrast to uniaxial tensile loading, compressive loads promote the formation of martensite variants that experience the maximum compression along the loading direction, which minimizes $G_{appl}$. Therefore, variant 1 is promoted to grow under uniaxial compressive loading conditions.

The microstructure obtained under triaxial compressive loading with σ = −100 MPa is shown in Figure 5. Figure 5 shows that unlike uniaxial tension and uniaxial compression, triaxial compressive loading does not result in different martensite microstructures. Instead, it resembles the martensite microstructure obtained without external loading. Comparing Figure 5 with Figure 2, it can be observed that the evolution process under triaxial compressive loading lags behind that under no external loading, and all variants are inhibited to varying degrees.

Depending on the martensite variants favored under different stress conditions, different stress states can lead to complex martensite microstructures. Martensitic phase transformations under loading conditions exhibit variant selection behavior, and this selection behavior is related to the magnitude and direction of external loading; the loading direction determines the types of variants that are favored, and the magnitude of the load influences the extent of variant selection. Due to the different contributions of external energy terms to the total system free energy, this also results in diversity in variant selection orientations during different loading processes. It can be said that applying external loads favors the formation of martensite variants that reduce the Gibbs free energy in the direction of the applied load, which is the common Magee effect [26,27].

### 4.3. Dynamics Analysis of Martensitic Phase Transformation under External Loading and Plastic Deformation

In order to visually observe the variation in martensite content under uniaxial loading, this study explores the effects of martensite variants and the total martensite volume fraction over time under uniaxial tensile and compressive loading. Figure 6 and Figure 7 illustrate the relationship between external loading, evolution time, and volume fraction.

Figure 6 and Figure 7 show that both martensite variants and the total volume fraction of martensite increase with increasing evolution time under both uniaxial tensile and compressive loading. Furthermore, the influence on volume fraction becomes more significant with increasing external loading. Under tensile loading, variant 1 decreases with increasing external loading, while variants 2 and 3 increase with increasing external loading. The opposite trend is observed under compressive loading. Therefore, as explained in the study by Yeddu et al. [28], externally applied loads may contribute to forming certain variants. It is worth noting that from Figure 7a, under compressive loading, variant 1 shows rapid growth after t* = 30. Under tensile loading, there is also a tendency for the rapid growth of variants 2 and 3, but it remains lower than the growth rate of variant 1 under compressive loading.

Figure 8 shows that, compared to no external loading, the martensite volume fractions under uniaxial tensile and compressive loading are ultimately greater. Therefore, uniaxial tensile and compressive loading both accelerate martensitic phase transformation. At the beginning of martensitic phase transformation, the volume fraction under uniaxial compressive loading is lower than that under uniaxial tensile loading. However, as the evolution progresses, it starts to increase rapidly after t* = 35 and surpasses the uniaxial tensile loading, ultimately resulting in a higher martensite volume fraction under uniaxial compressive loading. Traditional thermodynamics cannot reveal dynamic aspects of phase transformation, such as the evolution process. In this study, the phase field method reveals that uniaxial compressive loading accelerates martensitic phase transformation more effectively than uniaxial tensile loading from a dynamic perspective. This phenomenon is in excellent agreement with the experimental results of Hagiwara et al. [16].

In order to gain a deeper understanding of the microstructural evolution under a triaxial compression load, Figure 9 depicts the changes in the volume fractions of different martensitic variants and the total martensite with respect to the evolution time. From Figure 9a–c, it can be observed that the volume fractions of martensite and its variants increase as the evolution progresses. However, at the same time, the volume fractions of martensitic variants and the total martensite decrease with an increase in external loading. This simulation result is consistent with the research findings reported by Kakeshita [29] and other researchers. Therefore, under the influence of triaxial compression load, each martensitic variant is subjected to varying degrees of suppression.

To investigate the effects of plasticity and external stress on the elastic strain energy during the phase transformation process, the changes in elastic strain energy density during the martensitic transformation are shown in Figure 10.

In Figure 10, the green and blue lines represent the changes in elastic strain energy density during the microstructural evolution shown in Figure 2 and Figure 3, respectively. The red line represents the changes in elastic strain energy density without plastic accommodation and stress accommodation, where only self-accommodation effects regulate the elastic strain energy density. The results from the curves in Figure 10 indicate that plastic deformation and external tensile stress can reduce the elastic strain energy density, which is consistent with the conclusions obtained in the simulations from the referenced studies [19,30].

External loading and plastic effects both play significant roles in martensitic phase transformation. Figure 11 illustrates the relationship between martensite volume fraction and yield strength.

In Figure 11, “applied 30% strain” refers to uniaxial tensile deformation carried out during the simulation at a strain rate of 5.6 × 10^−4^ s^−1^, resulting in a final applied strain of 30%. Similarly, “applied 5% strain” indicates that uniaxial tensile deformation was conducted during the simulation at a strain rate of 5.6 × 10^−4^ s^−1^, leading to a final applied strain of 5%. During the phase transformation, yield strength is related to plastic deformation. When the yield strength is low, plastic effects are significant. Consequently, the strain energy relaxation due to lattice distortion is more pronounced, leading to a higher martensitic transformation temperature. As shown in Figure 11, the martensite volume fraction is minimally affected by changes in yield strength. In contrast, under the influence of various magnitudes of external loading, the martensite volume fraction exhibits a significant range of variation, spanning two orders of magnitude. This indicates that plastic effects have a minimal influence on the phase transformation process, and they can even be negligibly small under external loading conditions. Under the influence of external loading, the generation of martensite is primarily determined by the external load, with a minimal impact from plastic effects. This simulation result aligns with the experimental findings of Matsuoka and other researchers [31]. Therefore, in martensitic phase transformations under external loading, the external load plays a dominant role and has a greater influence on the transformation process than plastic effects. Previous studies mainly analyzed and discussed experimental results without delving deeply into the underlying physical mechanisms. This research employed a phase field simulation approach to provide theoretical insights into the effects of external loading and plasticity on martensitic phase transformations.

### 4.4. Prediction of Ms Temperature and TRIP Effect

The change in the martensite transformation temperature *M_s_* with respect to the applied uniaxial load is depicted in Figure 12.

From Figure 12, it can be observed that under tensile loading, the *M_s_* temperature increases with the applied uniaxial load. Under compressive loading, the *M_s_* temperature slightly decreases with increasing uniaxial load, with a relatively small effect on the *M_s_* temperature. Overall, in the material studied in this paper, uniaxial tensile loading increases the *M_s_* temperature and promotes martensitic transformation, while uniaxial compressive loading, although not significantly reducing the *M_s_* temperature, accelerates martensitic transformation. This is consistent with the results reported in the referenced studies [15,16]. The phase field simulations in this paper predict the phase transformation temperature, which not only validates the experimental results of previous researchers but also provides an explanation for the influence mechanism of uniaxial loading on martensitic transformation, revealing the connection between external loading and martensitic transformation at a fundamental physical level.

Figure 13 represents the variation in *M_s_* temperature with the magnitude of applied three-dimensional compressive loading. This work references experimental data studied by Patel and other researchers and conducts thermodynamic calculations based on the methods proposed by Patel and Cohen [15]. The red solid line and dashed line represent experimental values and thermodynamic calculation values, while the green scattered points represent the predicted values from the model in this paper. From Figure 13, it can be observed that the *M_s_* temperature decreases with increasing applied load. Based on the changes in volume fraction and *M_s_* temperature mentioned earlier, it can be concluded that three-dimensional compressive loading suppresses martensitic transformation. The prediction of *M_s_* temperature agrees with the results reported in the referenced literature [15,32].

Metals and alloys exhibit plastic deformation below the yield strength of the parent phase, leading to an increase in plasticity during the phase transformation process. This characteristic is known as Transformation-Induced Plasticity (TRIP) [33]. This study discusses the average equivalent plastic strain during the phase transformation process to investigate this phenomenon. The variation in average equivalent plastic strain with evolution time is shown in Figure 14.

Figure 14 indicates that as evolution progresses, the volume fraction of martensite continuously increases, leading to an increase in plastic strain. The higher degree of plastic strain occurs due to the martensitic phase transformation. Plastic deformation caused by temperature and applied stress changes does not occur during the simulated phase transformation process. Therefore, the martensitic phase transformation induces the plastic strain generated during the evolution. Furthermore, the externally applied stress is much lower than the yield strength of austenite, so the phase transformation process induces plasticity. The local yielding mechanism in this process occurs because the low stress induces the phase transformation, creating substantial internal stresses locally. At this point, the local internal stresses exceed the yield strength of austenite, resulting in local plastic deformation. As the phase transformation continues, the accumulation of local plastic deformation is manifested as macroscopic plastic deformation in the specimen.

## 5. Conclusions

(1)In the martensitic phase transformation process under external loading conditions, uniaxial tensile loading increases the martensitic phase transformation start temperature *M_s_* and promotes martensitic phase transformation. On the other hand, uniaxial compressive loading, while not significantly lowering *M_s_*, accelerates the transformation of austenite to martensite, resulting in an increase in martensite volume fraction and faster phase transformation kinetics. The acceleration effect of uniaxial compressive loading is superior to that of uniaxial tensile loading. Triaxial compressive loading reduces the *M_s_* temperature and inhibits the martensitic phase transformation. This conclusion aligns with relevant experimental findings.(2)The implementation of a phase-field model was employed to simulate martensitic phase transformation under the influence of both plastic accommodations and stress accommodations. The simulation results revealed that, in the presence of external loading conditions, the impact of external loading on martensite volume fraction outweighed the influence of plastic effects. Furthermore, external loading and plastic deformation can release the elastic strain energy during the martensitic phase transformation process.(3)Simulations of microstructure evolution during the martensitic phase transformation in Fe-Ni alloys were performed. The simulation results vividly demonstrate that different variants are favored to transform under corresponding stress conditions. Under uniaxial tensile loading, the growth of martensite variants 2 and 3 is promoted. Under uniaxial compressive loading, martensite variant 1 is favored to grow. Under triaxial compressive loading, the growth of all three martensite variants is inhibited.

## Figures and Tables

**Figure 1 materials-16-06849-f001:**
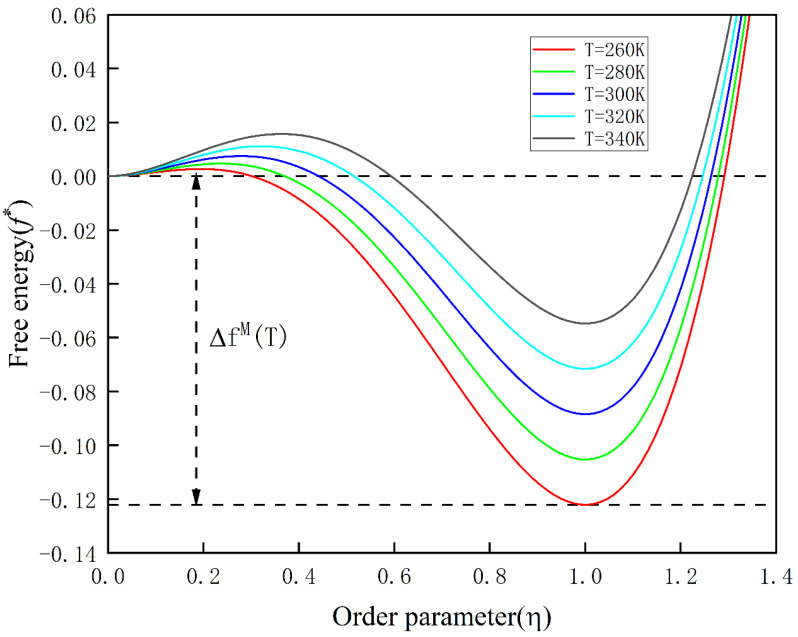
Schematic diagram of the local free energy function for displacement-type phase transformations at different temperatures.

**Figure 2 materials-16-06849-f002:**
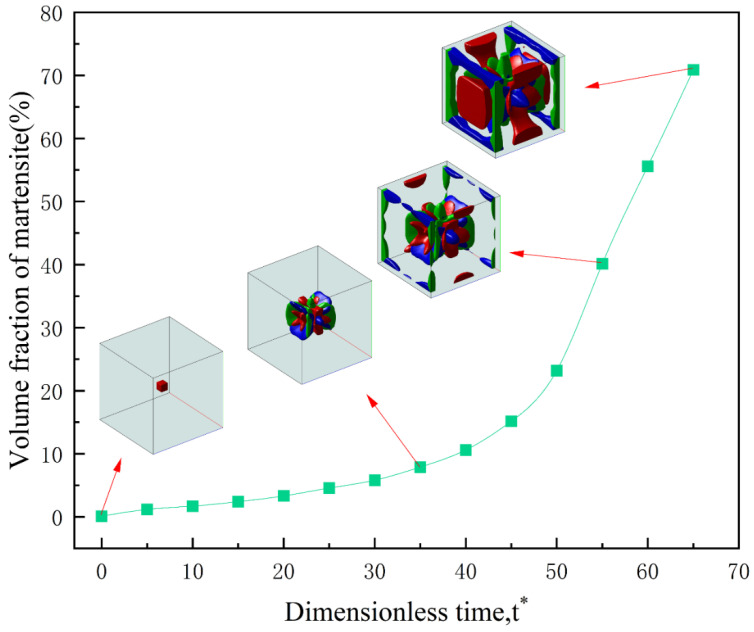
Variation in martensite volume fraction during phase transformation without loading.

**Figure 3 materials-16-06849-f003:**
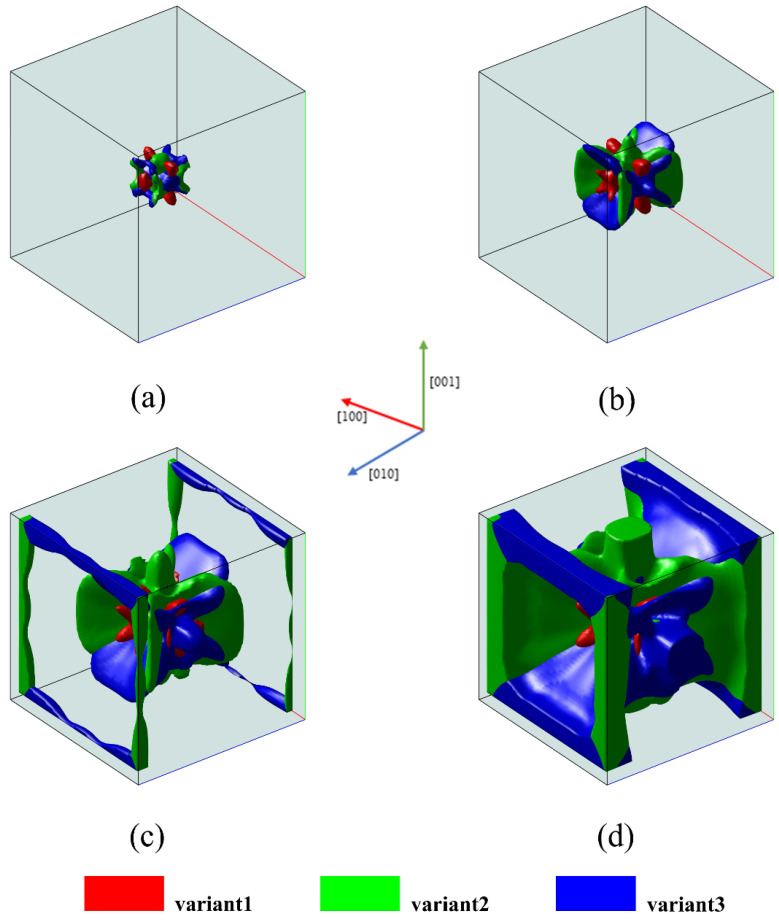
Three-dimensional microstructure evolution of martensitic variants under uniaxial tensile loading: (**a**) t* = 5; (**b**) t* = 25; (**c**) t* = 45; (**d**) t* = 60.

**Figure 4 materials-16-06849-f004:**
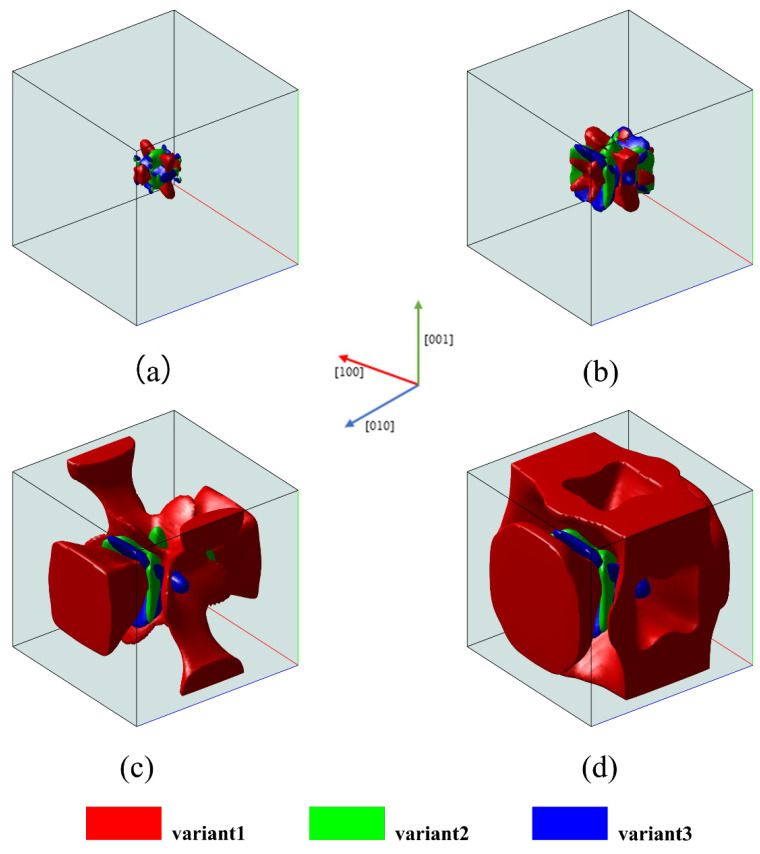
Three-dimensional microstructure evolution of martensitic variants under uniaxial compression loading: (**a**) t* = 5; (**b**) t* = 25; (**c**) t* = 45; (**d**) t* = 60.

**Figure 5 materials-16-06849-f005:**
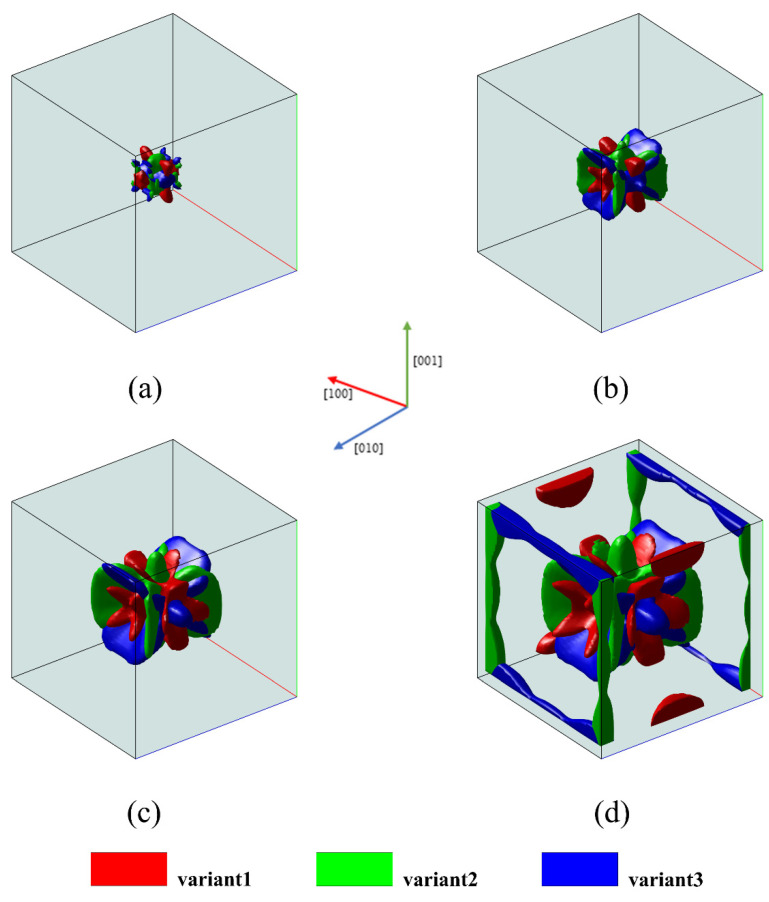
Three-dimensional microstructural evolution of martensite variants under triaxial compressive loading: (**a**) t* = 5; (**b**) t* = 25; (**c**) t* = 45; (**d**) t* = 60.

**Figure 6 materials-16-06849-f006:**
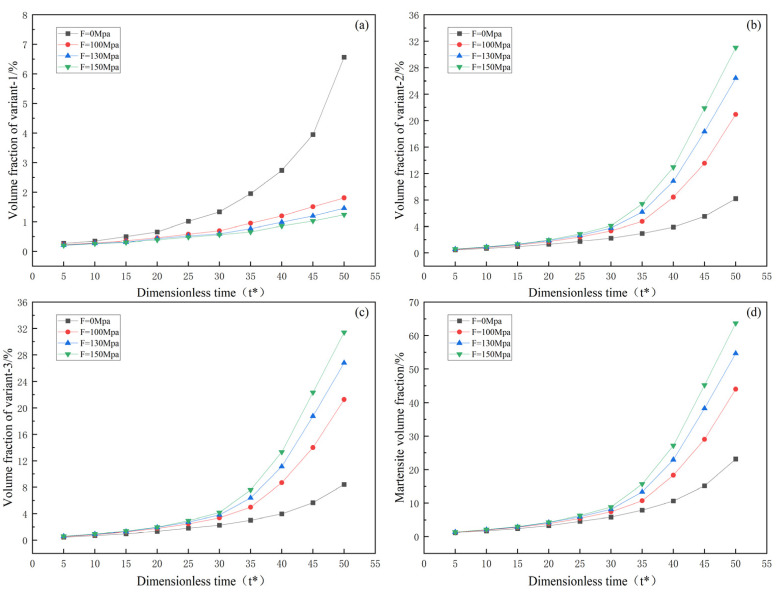
Evolution of martensite volume fraction with respect to time under uniaxial tension: (**a**–**c**) volume fractions of variants 1, 2, and 3; (**d**) total martensite volume fraction.

**Figure 7 materials-16-06849-f007:**
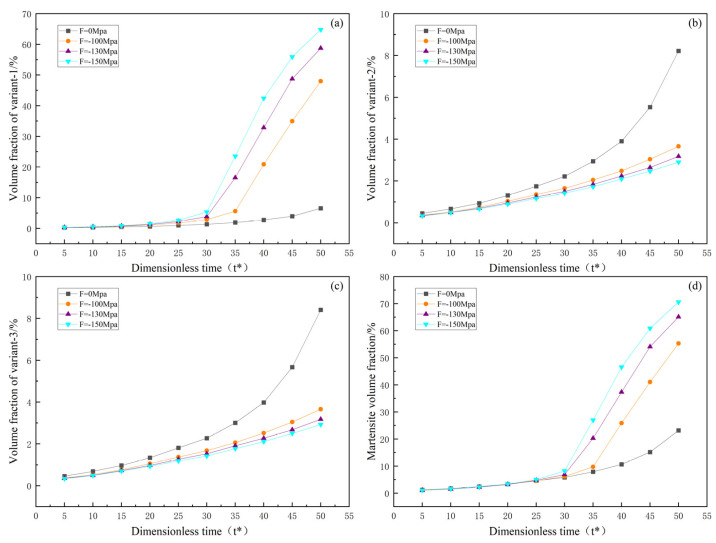
Evolution of martensite volume fraction with respect to time under uniaxial compression: (**a**–**c**) volume fractions of variants 1, 2, and 3; (**d**) total martensite volume fraction.

**Figure 8 materials-16-06849-f008:**
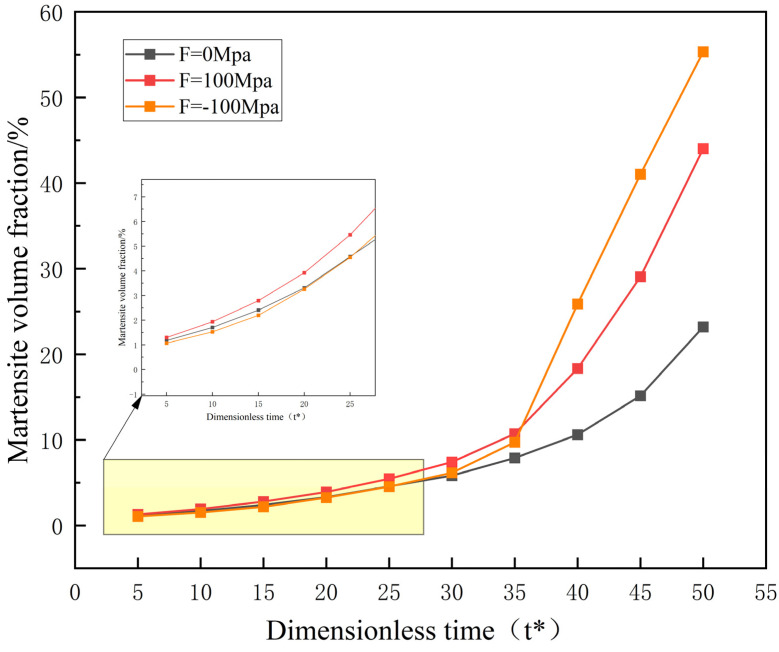
The evolution of martensite volume fraction under different applied uniaxial loads as a function of time.

**Figure 9 materials-16-06849-f009:**
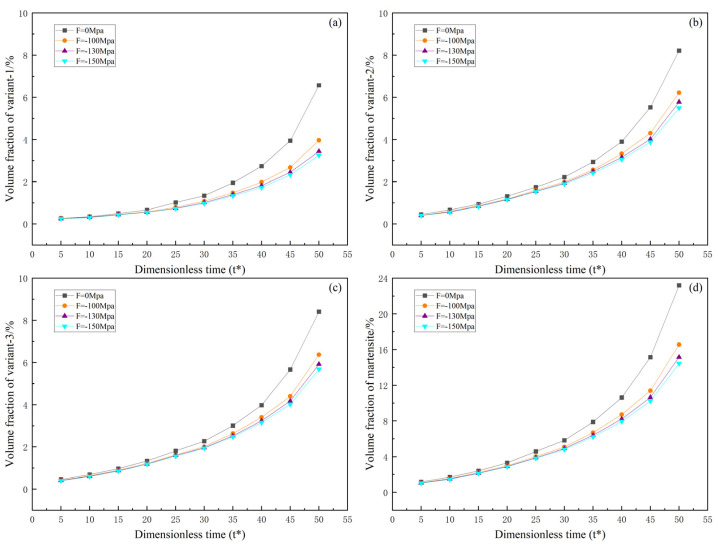
Evolution of martensite volume fraction with respect to time under triaxial compression loading: (**a**–**c**) volume fractions of variants 1, 2, and 3; (**d**) total martensite volume fraction.

**Figure 10 materials-16-06849-f010:**
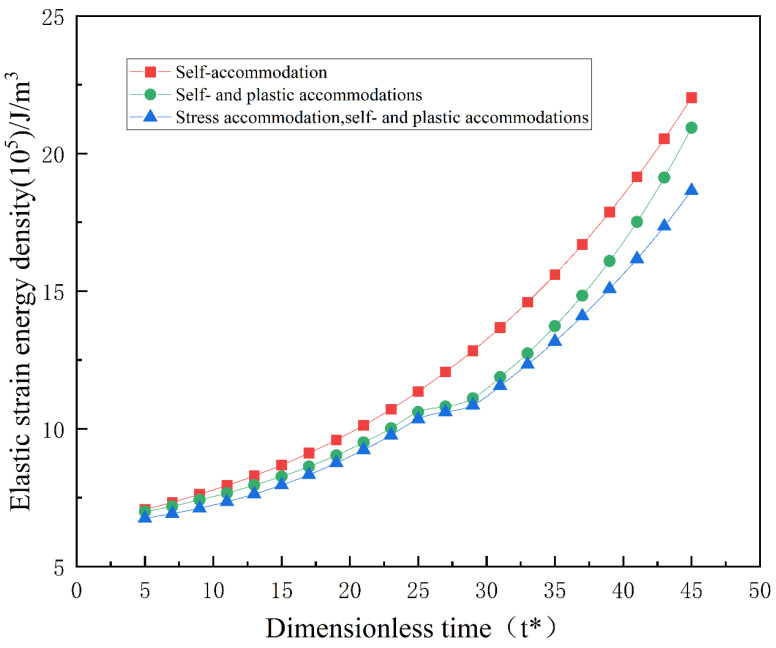
Elastic strain energy density during martensitic phase transformation under different conditions.

**Figure 11 materials-16-06849-f011:**
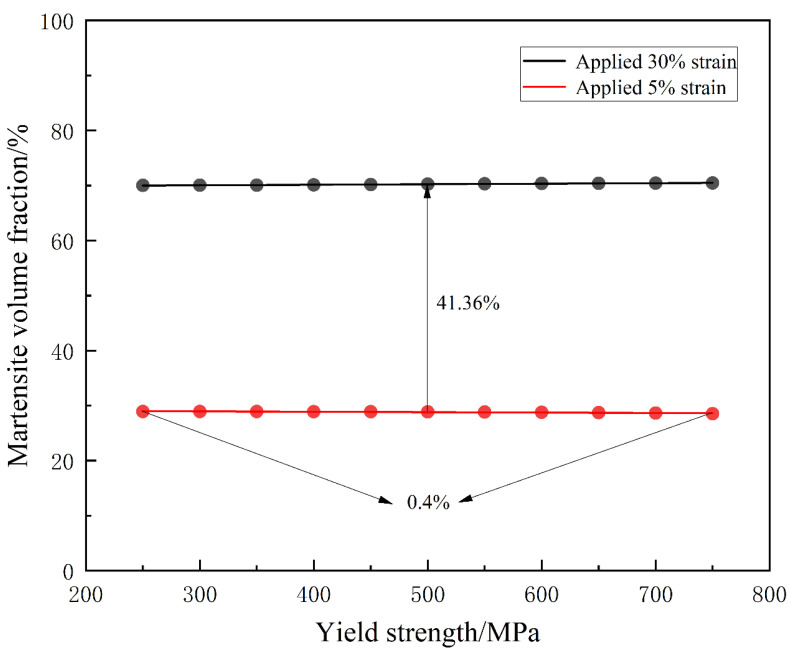
Effect of yield strength on martensite volume fraction under different external loading conditions.

**Figure 12 materials-16-06849-f012:**
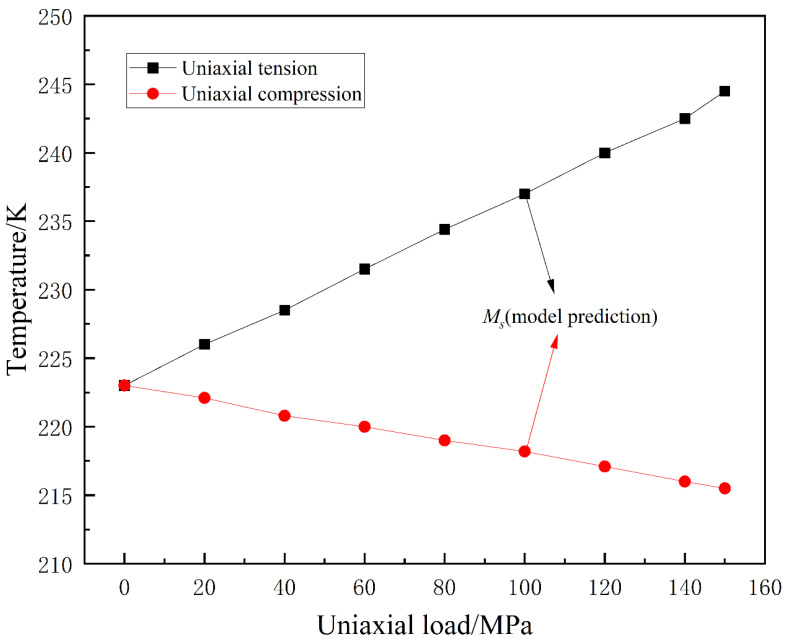
Variation in the martensite start temperature *M_s_* with externally applied uniaxial loading.

**Figure 13 materials-16-06849-f013:**
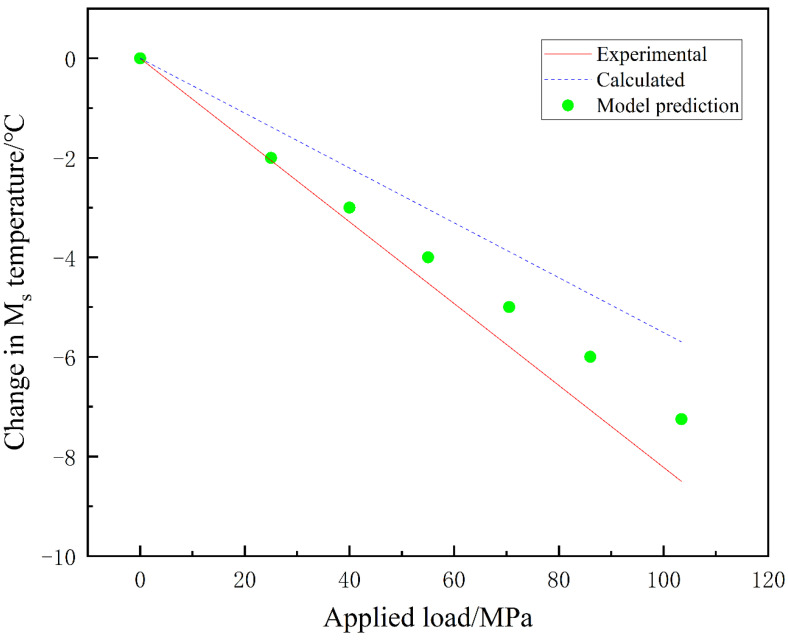
Relationship between the change in *M_s_* temperature and the triaxial compression load.

**Figure 14 materials-16-06849-f014:**
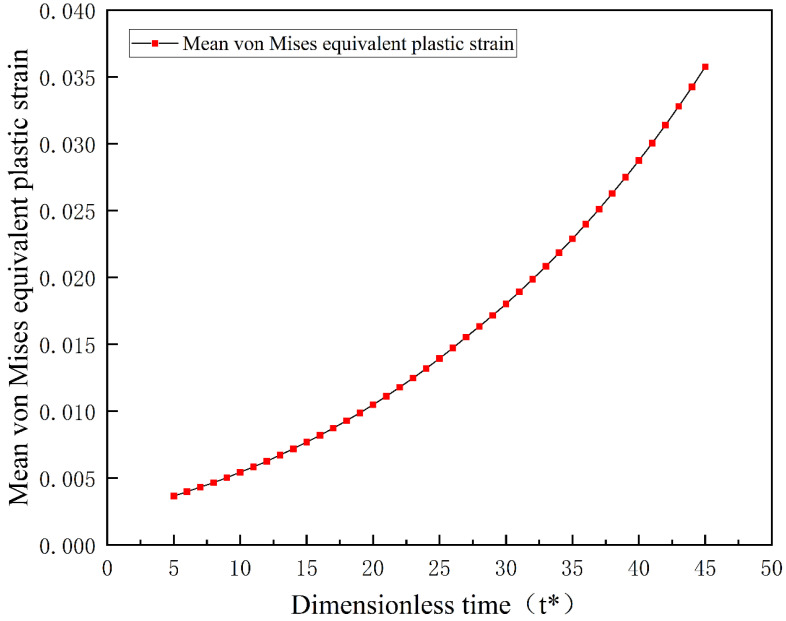
Variation in average equivalent plastic strain under uniaxial tensile load as a function of martensite volume fraction.

**Table 1 materials-16-06849-t001:** Physical parameters of Fe-Ni alloy.

Physical Parameter	Notation	Numerical Value
Latent heat for the transformation	*Q*/(J·m^−3^)	3.5 × 10^8^
Stress-free equilibrium temperature	*T_0_*/(K)	405
Shear modulus	*G*/(Pa)	2.8 × 10^10^
Martensite start temperature	*M_s_*/(K)	223
Characteristic energy	*E_0_*/(J·m^−3^)	1.026 × 10^7^
Poisson’s ratio	*v*	0.375
Bain strain tensors 1	$\varepsilon_{1}$	0.1322
Bain strain tensors 2	$\varepsilon_{3}$	−0.1994

**Table 2 materials-16-06849-t002:** Dimensionless parameters in the simulation.

Dimensionless Parameter	Numerical Value
$a_{0}^{*}$	60.02
$Q^{*}$	34.11
∆x^*^ = ∆y^*^ = ∆z^*^	1.0
∆t^*^	0.01

## Data Availability

The data that support the findings of this study are available from the authors upon reasonable request.

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
