# Peer review of "A Phase Field Study of the Influence of External Loading on the Dynamics of Martensitic Phase Transformation"

_materials, 2023, doi:10.3390/ma16216849_

Round 1
Reviewer 1 Report
Comments and Suggestions for Authors
The work submitted for evaluation titled: "A phase-field study of the influence of external loading on the dynamics of martensitic phase transformation" is an interesting theoretical study in which the authors presented the influence of external loading of the material on the kinetics of martensitic phase transformation. The work shows the influence of uniaxial tensile and compressive loads and the influence of triaxial compression of various intensity in the range of 100-150 MPa. They also considered the effect of plastic deformation.
I consider the presented introduction to the description of the problem being solved to be sufficient, and the description of the model is also sufficient (please check the last lower index in equation (3), in my opinion there should be the letter q here).
In Chapter 3, additional explanation is required for the adoption of the values Q* = 34.11 and a0* = 60.02. Please state on what basis the authors adopted such values? Is it possible to provide an approximate value of the real time corresponding to the dimensionless time assumed in the model?
In Chapter 4, Figure 6 requires improvement, it is illegible in the presented system, and for comparative purposes, the authors should show the total volume fraction of martensite in unloaded conditions in Figures 6d and 6h.
The conclusions presented by the authors result from the presented research results, and it is also interesting to compare model and experimental studies - however, the authors did not provide the conditions in which the experimental data were obtained and how the research was conducted, referring only to the literature [14] (line 395).
In the summary, the authors do not refer to the universality of the model, the tests were performed for an unspecified Fe-Ni alloy (only a reference to the literature). The authors should add a comment describing how universal the model is, what material parameters must be known and whether they are commonly available for other alloy systems (especially in terms of knowledge of the parameters a0 and Q).
In addition, the value of the work will be increased by information whether the model can have practical applications and in what practical situations its use can bring measurable technological benefits.
Reviewer 2 Report
Comments and Suggestions for Authors
Review report on "A phase-field study of the influence ox external loading on the dynamics of martensitic phase transformation"
In this manuscript, the authors report an effect of the stress field on the martensitic transformation in Fe-Ni modeled alloy by phase-filed simulation. It is interesting in the nucleation and growth of the martensite through the phase-field simulation. However, there are many reports on the effect of an external field on the martensitic transformation. Particularly, experimental works on the hydrostatic pressure effect on the kinetic of martensitic transformation have been reported by Kakeshita et al..
For example,
1. Effect of Hydrostatic Pressure on Martensitic Transformations in Fe–Ni and Fe–Ni–C Alloys, Trans. JIM 29(1988), 109. https://doi.org/10.2320/matertrans1960.29.109
2. Time-dependent nature of the athermal martensitic transformations in Fe-Ni alloys, Scripta Mater. 34(1996), 147. https://doi.org/10.1016/1359-6462(95)00483-1
3. Effect of Magnetic Field and Hydrostatic Pressure on Martensitic Transformation and Its Kinetics, Jpn. J. Appl. Phys. 36(1997), 7083. https://doi.org/10.1143/JJAP.36.7083
The reviewer recommends comparing the authors' work with the experimental study reported previously.
Besides, the authors discussed the kinetics of the martensitic transformation; however, why don't you discuss the works from the crystallographic and geometric viewpoints? The authors distinguish the martensite nucleated in the phase-field simulation as three variants. Since the phase-field crystal modeling has been recently reported, the reviewer suggests discussing the reason for the stress dependence (uniaxial and multi-axial) or preference for the nucleation and growth of martensite in more detail, if the authors can.
Reviewer 3 Report
Comments and Suggestions for Authors
A phase-field model was used for the modelling of isothermal martensitic transformation in FeNi alloy under external tensile and compressive stresses.
The results were compared with literal experimental results.
Remarks:
Directions of uniaxial loads are not declared relative to the austenite lattice. From the text it is revealed that the tensile stress was applied along the X axis (row 254), but this direction is not indicated on figs. 2-5. A schematic sketch about the austenite and the martensite variants unit cells would help greatly the better understanding.
In the text e.g. rows 244, 261 the load stress is denoted by F. It would be better to use the σ notation for the stress.
It is stated by the authors at row 15: ’ It is concluded that uniaxial loading promotes martensitic phase transformation, while triaxial compressive loading inhibits it.’ It is generally valid for all martensitic systems? In my opinion it would be better to restrict this statement for the present system.
The temperature parameter values are not reported. What T and T0 values were used at the simulations in the Gchem function? It is a fundamental parameter for the interpretation of results. Please, clarify the conditions necessary for the experimental checking of your simulations.
Plots on fig. 6 are very small (especially the labels). The arrangement of plots is not the best. Rearrangement of plots (a-d) and (e-h) in separate coulomns is recommended (if it is possible).
Misprint on labels Figs 9-11. instead of Mpa →MPa is suggested.
On fig 11 the dashed line shows thermodynamic calculation results of Ms stress dependence. What kind of thermodynamic calculation? (Clausius-Clapeyrone equation or something else?) Please, discuss it shortly.
Fig.9 is not discussed clearly. It is not clear what is the meaning of the applied 30% and 5% strains. Here more explanations are necessary.
Generally, it is necessary a more detailed description of the simulation conditions.
Reviewer 4 Report
Comments and Suggestions for Authors
The paper is well-motivated and well-performed continuing and extending
previous work of the authors. The aim, the method of analysis and the results are outlined correctly and comprehensively. The theoretical results are compared with experimental data and found to be in good agreement.
I found merely some – to my understanding – minor bugs cited below
- gibbs free energy-> Gibbs free energy
- Von Mises -> von Mises
I recommend acceptance for publication in the present form.
Minor editing of English language required
Round 2
Reviewer 2 Report
Comments and Suggestions for Authors
This revised manuscript is improved with the reviewers' comments. The authors' responses are also appropriate; the reviewer would like to suggest accepting the manuscript for publication in the Materials.